# Technical note: An assessment of the performance of statistical bias correction techniques for global chemistry-climate model surface ozone fields

Christoph Staehle[1], Harald E. Rieder[1], Arlene M. Fiore[2], Jordan L. Schnell[3,4]

[1]Institute of Meteorology and Climatology, BOKU University, Vienna, Austria

[2]Department of Earth, Atmospheric and Planetary Sciences, Massachusetts Institute of Technology, Cambridge, MA, USA

[3]Cooperative Institute for Research in Environmental Sciences, University of Colorado Boulder, CO, USA

[4] NOAA Global Systems Laboratory, Boulder, CO, USA

*Correspondence to:* christoph.staehle@boku.ac.at

**Abstract.** State of the art chemistry-climate models (CCMs) still show biases compared to ground-level ozone observations, illustrating remaining difficulties and challenges in the simulation of atmospheric processes governing ozone production and loss. Therefore, CCM output is frequently bias-corrected in studies seeking to explore the health or environmental impacts from changing air quality burdens. Here we assess four statistical bias correction techniques of varying complexity, and their application to surface ozone fields simulated with four CCMs, and evaluate their performance against gridded observations in the EU and US. We focus on two time periods (2005-2009 & 2010-2014), where the first period is used for development and training and the second to evaluate the performance of techniques when applied to model projections. We find that all methods are capable of significantly reducing the model bias. However, biases are lowest when we apply more complex approaches such as quantile-mapping and delta-functions. We also highlight the sensitivity of the correction techniques to individual CCM skill at reproducing the observed distributional change in surface ozone. Ensemble simulations available for one CCM indicate that model ozone biases are likely more sensitive to the process representation embedded in chemical mechanisms rather than to meteorology.

## 1 Introduction

Surface ozone ($O_3$) is both an air pollutant and greenhouse gas, formed in photochemical reactions involving precursor substances such as nitrogen oxides ($NO_x$) and volatile organic compounds (VOCs) of both anthropogenic and non-anthropogenic origin [e.g. Checa-Garcia et al., 2018; Lelieveld & Dentener, 2000; Monks et al., 2015]. In addition to the availability of precursor gasses, the $NO_x$ to VOC ratio as well as solar radiation and ambient air temperature, controlling emissions of biogenic VOCs (BVOCs) and chemical reaction rates, play a crucial role for $O_3$ formation [Chameides et al., 1988; Sillman, 1999; Sillman et al., 1990]. Tropospheric $O_3$ abundance is also substantially influenced by stratospheric intrusions, which can, in certain regions or during specific events, alter concentrations significantly [Akritidis et al., 2010; Lin et al., 2015; Stohl et al., 2003]. $O_3$ is associated with a variety of detrimental human health effects, especially in the context of the respiratory and cardiovascular system, resulting in about 5-20 % of premature deaths attributable to ambient air pollution [Gu et al., 2023; Malashock et al., 2022; Monks et al., 2015; Murray et al., 2020; Pozzer et al., 2023; Zhang et al., 2019]. In addition to its negative health effects, $O_3$ can compromise the metabolism of plants through stomatal uptake and cause damage to leaf surfaces, thereby affecting biomass and crop production [Da et al., 2022; EEA, 2020; Fleming et

al., 2018; Mills et al., 2018; Monks et al., 2015]. Consequently, a large body of studies examine past, present and future development of surface $O_3$ burdens as well as resulting health and ecological impacts on both regional and global scale [e.g. Da et al., 2022; Meehl et al., 2018; Nolte et al., 2018; Westervelt et al., 2019].

Studies exploring future changes in surface $O_3$ burdens and their implications for human health and the biosphere rely on simulated fields of chemistry-climate models (CCMs) and chemistry-transport models (CTMs). However, despite ongoing development, these models show deficiencies in the adequate representation of ground-level $O_3$ on regional to local scale and changes therein when compared to observations [e.g. Griffiths et al., 2021; Karlický et al., 2024; Turnock et al., 2020; Young et al., 2018]. This shortcoming raises questions regarding the reliability of the simulated surface ozone response to changes in precursors and ambient climate. The number of possible reasons for the deviation of model output and observations increases with the complexity of the models. However, the published literature commonly suggests issues with emissions fed into the models, the applied chemical mechanism, meteorology and deposition in addition to uncertainties associated with the spatial resolution [e.g. Archibald et al., 2020; Liu et al., 2022; Young et al., 2018]. To overcome these issues, also as individual experiments are computationally expensive similar to climate studies, statistical bias correction techniques of different complexity are frequently applied to correct global model fields. Such corrections allow the diagnosis of changes in ambient meteorological conditions and ozone in isolation or combination and to investigate related impacts on human health. Machine learning approaches are increasingly being used for correction purposes [e.g. Liu et al., 2022]. These methods, however, usually have the disadvantage of behaving like a "black box", i.e. algorithms lack traceability and thus physical insights as to the root cause of biases. To date no detailed comparison of different statistical bias correction techniques for surface ozone burdens has been performed and the present study aims to close this gap.

Here we analyze historical simulations from 3 different global CCMs contributing to the Coupled Model Intercomparison Project phase 6 (CMIP6), as well as a 13-member ensemble of the CESM2-WACCM6 model for the European (EU) and contiguous United States of America (US) domain. For an assessment of model performance, we compare model outputs with gridded observational datasets available for both domains. First, we evaluate the ozone fields of the individual CCMs with observations and contrast the magnitude, sign and seasonality of the bias among CCMs. Thereafter, we apply a set of statistical bias correction techniques aiming for a reduction of the initial bias, independent of its origin, and evaluate the performance of these methods to identify if a particular correction technique is preferable across models.

Since the model simulations are "free-running", and thus create their own meteorology internally, a direct day-to-day comparison with the observations is not meaningful. Hence, our analysis primarily aims to evaluate the distribution of the $O_3$ fields in a statistical sense. Given the importance of ozone for human health we focus on the upper tail of the maximum daily 8-hour average (MDA8) $O_3$ distribution, and the frequency of occurrence of exceedance of health-related target values for Europe and the US.

## 2. Data & Methods

### 2.1 Model and observational data

The $O_3$ data sets explored in our analysis are hourly surface $O_3$ outputs from three CCMs (GFDL-ESM4, UKESM1-0-LL and EC-Earth3) contributing to CMIP6, and a 13-member ensemble simulation created with CESM2-WACCM6. For most of our study, we use only the first ensemble member of CESM2-WACCM6 in analogy to the other CCMs, given the overall

heterogeneity in the number of members available per model. In section 3.4, we focus on the chemical vs. meteorological driving of model biases and utilize the entire CESM2-WACCM6 ensemble. We also obtain observed MDA8 $O_3$ with a spatial resolution of 1° x 1° per grid cell for both the European and the US domain using an extended dataset constructed using the methods of Schnell et al. [2014]; [2015] and Schnell and Prather [2017], one which was designed specifically to compare against gridded CCMs. The dataset is constructed using an inverse distance weighted interpolation method that includes a declustering component similar to kriging; i.e., clustered (within 100 km) observations' weights are reduced such that those stations (often located around urban centers) are not disproportionately used in the interpolation. For the US domain, point based observations that are used in the interpolation include the US EPA's Air Quality System (AQS), the US EPA Clean Air Status and Trends Network (CASTNET), and Environment Canada's National Air Pollution Surveillance Program (NAPS); for the European Domain we include the EMEP and European Environment Agency's AirBase network (excluding stations designated as traffic). The exponent for the distance component is 2.5 and a maximum distance of 500 km is used for the weights. Parameters were estimated using a leave-N-out cross-validation technique. Estimations are made at 25 equally spaced points within each 1° x 1° cell and trapezoidally averaged. Other recent work has used this extended dataset [e.g., Ducker et al., 2018; Garrido-Perez et al., 2019; Guo et al., 2018]. Schnell et al. [2014] estimated an RMSE of 6-9 ppb for individual stations and 0-3 ppb for the grid cell averages; Ducker et al. [2018] estimated a mean bias of 5-10 ppb with the updated dataset over their study locations. For the analysis here the interpolation is performed on hourly abundances and the MDA8 $O_3$ is estimated using the interpolated hourly fields. Note, we apply the nomenclature of the European Union for the calculation of the MDA8 $O_3$ values in both domains, i.e. the eight hour average for a given hour is derived using the data of that specific hour and the preceding seven hours [EUR-LEX, 2008]. For convenience, the data is provided on a public repository, see data statement below.

To allow for an optimal comparison, the model data is regridded using an ordinary inverse distance weighting algorithm to match the spatial extent of the observations. In addition, all data sets are harmonized regarding their temporal resolution by removal of days not included in any of the other datasets, resulting in a 358-day calendar (30 days per month except for February). MDA8 $O_3$ is derived for each dataset and time step according to the European nomenclature as mentioned above. For the historical analysis we use 2005 to 2009 to evaluate the baseline bias of the individual CCMs and establish the performance of individual bias correction techniques. The time slice 2010 to 2014 is used subsequently, to evaluate the performance of our methods for model projections.

### 2.2 Bias correction methods

For statistical bias correction we apply four different techniques, which are detailed below. Here $M_q$ and $O_q$ denote quantiles ($q \in 1, ..., N \mid 1 = min, N = max$) of the model and observational distributions, respectively. The running index $j$ marks individual MDA8 $O_3$ model values. Additionally, we use the indices $hist$ and $proj$ to differentiate between historical and projected data. Primed terms indicate the bias corrected model outputs.

### 2.2.1 Mean bias correction (MB)

$$\overline{\Delta M} = \frac{1}{N} \sum \left( M_q^{hist} - O_q^{hist} \right) \tag{1}$$
$$= \overline{M_q^{hist}} - \overline{O_q^{hist}}$$
$$M_q'^{proj} = M_q^{proj} - \overline{\Delta M} \; ; \; M_q'^{hist} = M_q^{hist} - \overline{\Delta M} \tag{2}$$

The MB is a commonly used approach assuming a constant offset between model and observations. As an initial step we derive the average difference of the historical model and observational percentiles (alternatively the difference between the mean values of both empirical cumulative distribution functions (ECDFs) can be computed). Subsequently we subtract the result of Eq. (1) from each quantile of the projected model distribution to retrieve a bias corrected model ECDF (Eq. (2)).

### 2.2.2 Relative bias correction (RB)

$$\bar{c} = \frac{1}{N} \sum \left( \frac{M_q^{hist} - O_q^{hist}}{O_q^{hist}} \right) \tag{3}$$

$$M_q'^{proj} = M_q^{proj} - \bar{c} * O_q^{hist} \tag{4}$$

Here, similar to the MB method, we assume that model and observations differ by a constant factor. In contrast to the MB correction, however, we derive the average of the relative deviation of the historic model and observational percentiles (Eq. (3)). The bias corrected model projection (Eq. (4)) is then calculated as the difference between the raw model and the observed quantiles times the correction term established in Eq. (3).

### 2.2.3 Delta correction (DC)

$$\Delta M_q = M_q^{proj} - M_q^{hist} \tag{5}$$

$$M_q'^{proj} = O_q^{hist} + \Delta M_q \tag{6}$$

The DC approach follows the methodology detailed in Rieder et al. [2018]. In contrast to the MB and RB methods it is assumed that, while the individual model values may be biased, the system response (i.e., change between two time periods) is represented adequately by the model. Therefore the deviation between future and base period model data is calculated for all quantiles individually (Eq. (5)). Finally the corrected model projection is derived as the observed distribution plus the initially computed model change (Eq. (6)).

### 2.2.4 Quantile mapping (QM)

$$R_q^{hist} = \frac{O_{q+1}^{hist} - O_q^{hist}}{M_{q+1}^{hist} - M_q^{hist}} \tag{7}$$

$$c_j^{hist} = R_q^{hist} * \left( M_j - M_q \right) ; M_q \leq M_j < M_{q+1} \tag{8}$$

$$M_j'^{hist} = O_q^{hist} + c_j^{hist} \tag{9}$$

$$\Delta M_q = M_q^{proj} - M_q^{hist} \tag{10}$$

$$m_q'^{proj} = M_q'^{hist} + \Delta M_q \tag{11}$$

$$R_q^{proj} = \frac{m_{q+1}'^{proj} - m_q'^{proj}}{M_{q+1}^{proj} - M_q^{proj}} \tag{12}$$

$$c_j^{fut} = R_q^{proj} * \left( M_j^{proj} - M_q^{proj} \right) ; M_q^{proj} \leq M_j^{proj} < M_{q+1}^{proj} \tag{13}$$

$$M_j'^{proj} = m_q'^{proj} + c_j^{proj} \tag{14}$$

The term "quantile mapping" summarizes a variety of similar bias correction approaches used within the climate research community [e.g. Lehner et al., 2023]. Here, however, we follow the method described for CCM outputs in Rieder et al. [2015]. In contrast to the other methods used in this study, the QM is a multi-step approach. The first steps, illustrated in Eq. (7) to (9), consist of the computation of a bias corrected historic model distribution. Next, the result is used to create a bias corrected future ECDF, similar to the DC method (Eq. (10) and (11)), which is then employed to derive the bias corrected

future model data (Eq. (12) to (14)). In contrast to Rieder et al. [2015] however, who suggested a fixed apportionment for the quantiles to avoid non-meaningful results by executing undefined operations, especially in Eq. (7) and (12), (i.e. denominator equals zero or both denominator and numerator equal zero), we employ here a variable algorithm to select the optimal number of percentiles for each individual realization of the QM method. This is achieved by fixing the minimum and maximum values of the model ECDF and allowing for all quantiles with unique values within this range, i.e., if several

quantiles share the same value, which might be the case, especially for narrow distributions, only the first quantile is used.

    All four methods are applied to the ECDFs of the individual CCM datasets (1) on a monthly basis within the base time interval, (2) for each grid cell individually (in contrast to Rieder et al. [2015], who used a regional approach), (3) for both the EU and US domains. While it is implied that the model data differs from the observations by a constant factor for the

MB and RB methods, the DC and QM techniques assume that the difference between the future and reference period is represented adequately in the individual models, independent of the prevailing model bias. In contrast to the QM method, which provides the opportunity to directly correct individual daily MDA8 $O_3$ values, the application of the MB, RB (according to the methodology detailed above) and DC techniques solely results in new model ECDFs. The mapping algorithm, detailed in Eq. (7) to (9) is therefore applied further to the outputs of these three correction methods. Thereby the

model data is mapped onto the bias corrected ECDFs, allowing for an optimal comparison of original- and bias corrected model data with the observations and the results from the other correction techniques under investigation here.

    To quantify the initial biases as well as the remaining bias after application of the individual correction techniques we derive the number of days above the target value for the protection of human health (120 μg/m$^3$ in the EU (approximately 60 ppb)

and 70 ppb in the US) and the residual bias of the ECDFs on seasonal and annual time scales [EPA, 2015; EUR-LEX, 2008, 2011].

## 3 Results

### 3.1 Model evaluation

    We start by evaluating the performance of the global models in representing the MDA8 $O_3$ burden for the historical time

period (2005-2009). Fig. 1a,b shows the pooled MDA8 $O_3$ probability density function for models and gridded observations for the EU and US domains. Pronounced differences emerge between the individual models and observations for both domains. Generally, the models are biased high compared to observations, and the amplitude of the bias varies substantially among models. One exception in this regard is the EC-Earth3 model, which shows a high bias compared to observations across the majority of the MDA8 $O_3$ distribution but in contrast to other models a low bias at the upper tail.

We further investigate the magnitude of the model biases in Fig. 1c,d by contrasting the annual average number (and seasonal partitioning) of days above the target value to protect human health, defined as 60 and 70 ppb for the EU and US

domains, respectively. For the observations we find a domain average number of exceedance days of the target values of 8 (five for summer and 3 for spring) and 3 (2 for summer and 1 for spring) days for the EU and US domains in 2005-2009. While the models agree with observations regarding a preferred occurrence of non-attainment days in summer, all models but EC-Earth3 substantially overestimate the occurrence frequency of exceedance days. The domain average bias in non-attainment days for the EU ranges between 5 days in EC-Earth3 and 113 days in UKESM1-0-LL. In contrast, values for the US vary between 2 to 79 days. Overall our findings indicate a slightly better agreement of CCMs regarding the policy relevant metrics in the US than EU, which has to be taken with caution given also the regional difference in the MDA8 $O_3$ target value. Assuming the same target threshold as for Europe, we find that the number of exceedance days ranges between 20 and 174. Table 1 provides a summary of the occurrence frequency of MDA8 $O_3$ extremes for models and observations on an annual and seasonal basis (note, fall and winter are grouped together (FW) due to the small number of exceedance days derived for these seasons).

| 2005-2009 | | | | | |
|---|---|---|---|---|---|
| | **Obs** | **EC-Earth3** | **CESM2-WACCM6** | **GFDL-ESM4** | **UKESM1-0-LL** |
| **MAM** | 3 (1; 7) | 1 (0; 4) | 4 (2; 17) | 4 (2; 10) | 43 (18; 63) |
| **JJA** | 5 (2; 12) | 4 (2; 12) | 13 (10; 29) | 27 (18; 42) | 57 (51; 78) |
| **FW** | 0 (0; 2) | 0 (0; 4) | 1 (1; 5) | 4 (2; 8) | 13 (10; 33) |
| **annual** | 8 (3; 21) | 5 (2; 20) | 18 (13; 51) | 35 (22; 60) | 113 (79; 174) |
| 2010-2014 | | | | | |
| **MAM** | 1 (0; 5) | 0 (0; 2) | 3 (1; 15) | 2 (1; 6) | 39 (10; 58) |
| **JJA** | 3 (1; 8) | 1 (1; 6) | 9 (7; 24) | 29 (17; 40) | 54 (37; 72) |
| **FW** | 0 (0; 2) | 0 (0; 2) | 1 (1; 6) | 4 (1; 8) | 14 (6; 27) |
| **annual** | 4 (1; 15) | 1 (1; 10) | 13 (9; 45) | 35 (19; 54) | 107 (53; 157) |

Table 1: Average number of exeedance days (i.e., the number of days above the target threshold of 60 ppb (EU) and 70 ppb (US), respectively) per grid cell derived from observations and individual raw model data for the EU and US (given in parenthesis) for spring (MAM), summer (JJA), fall and winter (FW) and annual. Note numbers in italics in the parentheses are derived applying the EU threshold for the US.

Next, we turn to model biases in the spatial domain. Figure 2 shows the difference in the average number of days above the target value for individual models to observations (note, grey shaded areas indicate a marginal difference of up to ± two days). The spatial distribution of differences confirms the biases detailed above, showing regionally varying but distinct biases of the models examined. Of the models examined, the EC-Earth3 model performs best in both domains with a domain average bias of +7 (EU) and +3 days (US), respectively. While pronounced differences in the magnitude of the bias between individual models occur, the spatial patterns in bias are quite similar. In particular a north to south gradient emerges in the European domain with significantly higher model biases in the Mediterranean region and small to negligible biases in Scandinavia and the UK. For the US we find across models less pronounced biases in the Midwest, while substantial biases emerge in the North- and Southeast and Southwest.

To investigate the consistency of the spatial bias in models compared to observations we expand the analysis to the 2010-2014 time period (Fig. S1 & S2). Although slight variations are found for individual seasons, overall the result for this time

period resembles those obtained for 2005-2009 in both the US and EU domain (see Fig. S1). This result provides further confidence in the robustness of our assessment of general model biases in the MDA8 $O_3$ distribution and the modelled frequency of non-attainment days.

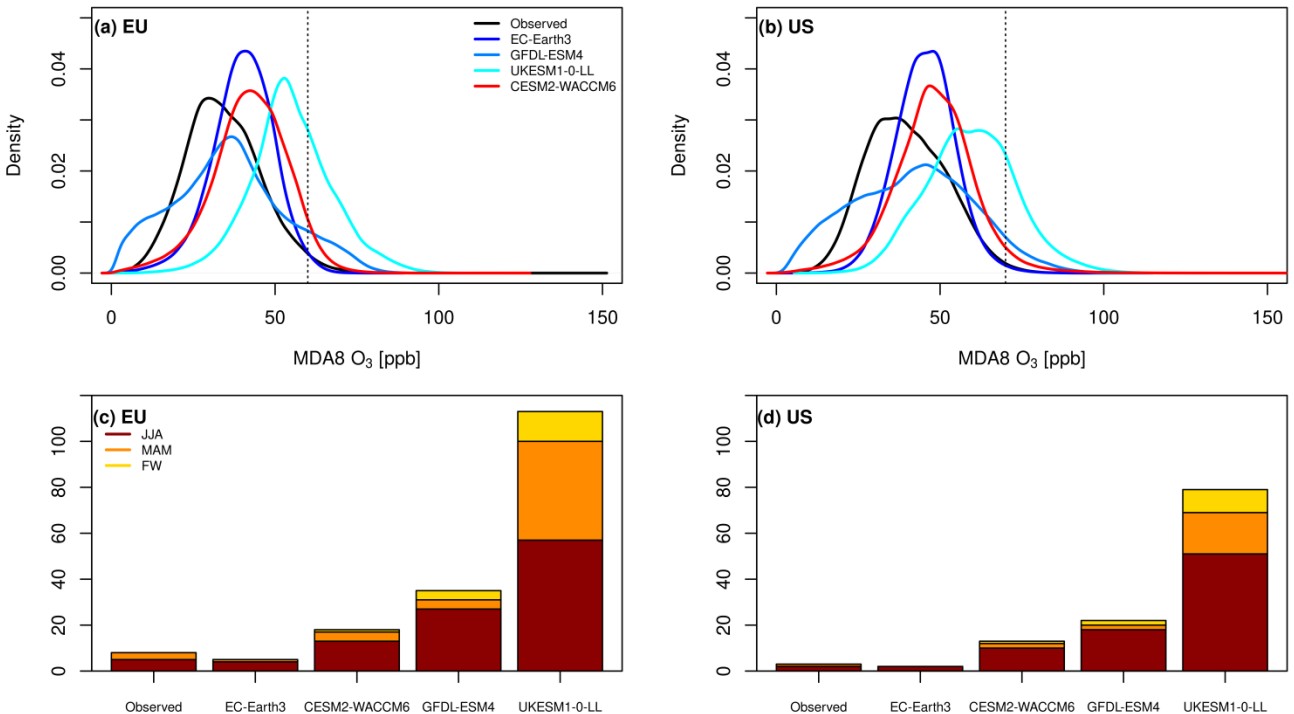

**Fig. 1:** Probability density function (PDF) of observed (black) and modeled (coloured) MDA8 $O_3$ during 2005 to 2009 in the EU (a) and US (b) domain. Average number of days above the MDA8 $O_3$ target value per grid cell for summer (JJA – red), spring (MAM – orange) as well as fall and winter months (FW – yellow) during 2005 to 2009 in the EU (c) and US (d) domains. In panels (a) and (b) dashed vertical lines indicate the target value for the protection of human health. The annual average number of exceedance days in (c) to (d) is given by the sum of the individual segments, i.e. the total height of the bars.

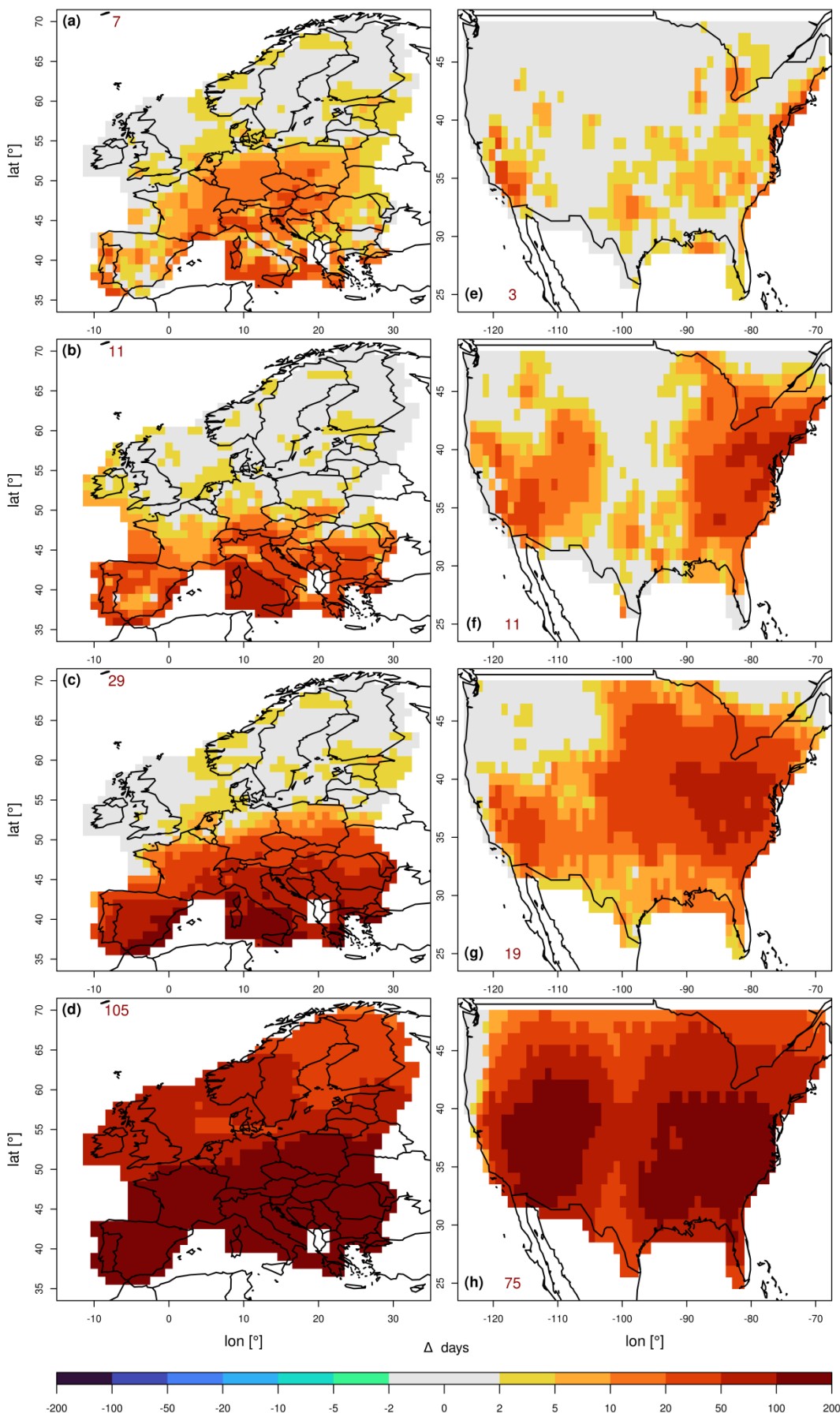

**Fig. 2:** Difference in the average number of days above the MDA8 O$_3$ target value in CCM simulations (EC-Earth3 a & e, CESM2-WACCM6 b & f, GFDL-ESM4 c & g, UKESM1-0-LL d & h) compared to gridded observations for the European (left) and US domain (right). All panels show differences during 2005-2009. Red numbers in the upper/lower left corner indicate the grid cell average anomaly. Grey shading indicates differences within $\pm$ 2 exceedance days.

### 3.2 Bias correction for the base period 2005-2009

Having illustrated the model biases for the past, we turn next to bias correction. To this end we apply the individual bias correction methods to model outputs for 2005-2009 and evaluate their performance for the MDA8O$_3$ distribution and the number of non-attainment days. The DC method represents an exception in this case as applying this method, by definition, would yield a "perfect" agreement with the observational ECDF. Accordingly, any potential deviations from observed ECDF would be a mere result of uncertainties associated with implementation, in particular the mapping algorithm and rounding, and thereby do not represent the performance of the DC method in context of the base dataset. The performance of the DC method will be however, assessed, along with the other methods when applied to the evaluation period 2010-2014.

Figure 3 shows the distribution of the grid-cell level bias in the number of exceedance days for the European (a – d) and US (e – h) domains. All methods reduce the bias substantially. The MB and RB methods yield similar results. Both methods tend to overcorrect the bias, yielding residual biases for individual grid-cells varying between -22 and +8 days (EU) and -10 to +6 days (US), with MB performing slightly better. In contrast, the QM method yields an almost perfect agreement (comparable to the DC method as detailed above) with observations. Residual biases are between -2 and +1 days for Europe and 0 days for the US.

Spatial distributions of the anomaly on exceedance days are illustrated in Fig. S3 and S4. We find that the application of a particular method yields similar spatial patterns of improvement independent of the model to which it is applied and of the initial model bias. For the MB and RB approaches, the spatial gradient in the bias identified in the raw models remains for the EU domain, although with reversed sign for the majority of applications, i.e., stronger overcorrection in the Central Europe and the Mediterranean than in the northern parts of EU domain. For the US the MB and RB methods perform better compared to Europe. This finding, however, is attributable to the higher target threshold rather than the actual performance of these methods as shown in section 3.1. The QM method best captures the observations in both domains.

We examine the PDFs of the bias-corrected model data for conformity with the observations (see Fig. S5). While all correction methods lower the bias across the whole distribution, the MB and RB approaches still deviate from the observations. In contrast, the distribution of the QM-corrected data is almost perfectly aligned with the observational PDF, independent of the model and domain. In summary, our evaluation for the baseline period indicates a clear preference for the QM (or the DC) method.

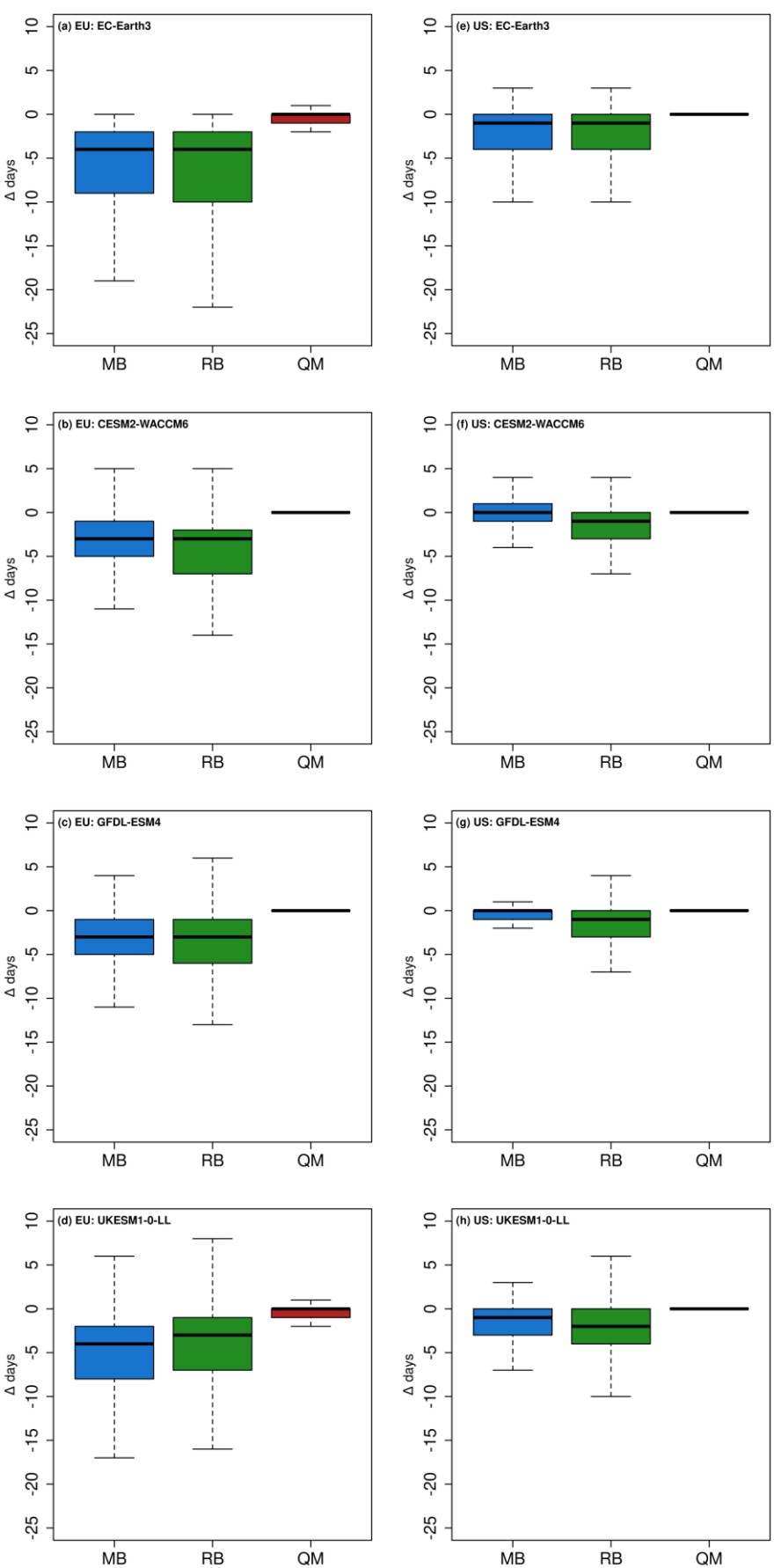

**Fig. 3:** Boxplots of the average residual bias in exceedance days pooled across grid cells in the individual CCMs in 2005-2009: (a) EC-Earth3, (b) CESM2-WACCM6, (c) GFDL-ESM4 and (d) UKESM1-0-LL models in the EU domain, (e)-(h) as (a)-(d) but for the US domain. Blue, Green and red colour indicate the MB, RB and QM correction methods respectively.

### 3.3 Bias correction performance in the evaluation period 2010-2014

Next, we turn the focus to the results obtained with individual bias correction techniques during the evaluation time period (2010-2014). We apply the adjustment methods to the MDA8 $O_3$ outputs of the individual models, but treat the data as independent realizations, in order to assess the methods performance for their applicability to future projections (see section 2).

Figure 4 shows the distribution of the residual grid-cell level mean bias compared to observations for the number of exceedance days of the target value. Here we find a larger residual bias, ranging between -17 and +11 exceedance days in the European domain (Fig. 4a-d) than in the US (Fig. 4e-h) where the bias after correction varies between -5 to +5 days across grid cells. Furthermore, contrasting the performance of the individual bias correction techniques yields a curious result, as we no longer identify an individual correction technique as optimal across models and spatial domains.

We further explore the spatial distribution of the residual bias. Compared to the base period the MB and RB (see Fig. S6-S7 a-d & e-h) corrected models show an improved agreement compared to observations. While the spatial patterns of bias distributions are similar to the 2005-2009 period (except for the GFDL-ESM4 model) an improvement compared to the base period is found for the northern and eastern European countries as well as the south-east US. The residual bias worsens in the central EU and the Mediterranean as well as the southwest US when applied to the GFDL-ESM4 model. For the DC and QM approaches (see Fig. S6-S7 i-l & m-p) on the other hand, we find a significantly increased residual bias (of both positive and negative sign), independent of model and domain.

Although all methods applied are still capable of significantly reducing the bias, these results, in contrast to those for the base period, no longer allow the identification of a sole ideal correction method, indicating changes in the underlying processes contributing to the bias. Our findings show that the correction approach yielding the lowest residual bias varies strongly across models and spatial domain. For example, while the QM method performs best for the CESM2-WACCM6 in the EU domain (Fig. 4b) the RB method yields a smaller residual bias in the US (Fig. 4f).

These results are supported by the analysis of the PDFs of the bias corrected model output (Fig. S8). While the conformity with observations remains widely similar for the majority of the distribution, the adjustment of the high tail yields slightly better results in context of the MB and RB methods, when compared to the base period. Contrarily, the distributions of both the DC and QM methods show a good agreement with the low tail and the midsection of the observational PDF. The performance however, deteriorates towards the high tail, partially resulting in an overestimation of the monitored distribution, especially in the European domain.

To further investigate this curious result we examine, on a quantile basis across the MDA8 $O_3$ distributions, i) the error resulting from the initial bias correction of the base period ($E_B$) and ii) the error resulting from the deviation of the models change between the base and evaluation period when compared to observations ($E_\Delta$).

$$E_B = M_q'^{hist} - O_q^{hist} \tag{15}$$

$$E_F = M_q'^{proj} - O_q^{proj} \tag{16}$$

$$= \left( M_q'^{hist} + \Delta M_q \right) - \left( O_q^{hist} + \Delta O_q \right)$$

$$= E_B + \Delta M_q - \Delta O_q = E_B + E_\Delta$$

The results of this analysis are exemplarily shown for the CESM2-WACCM6 model in Fig. 5 for the EU (a-c) and the US (d-f) (note the illustrations for the other models are included in supplemental Fig. S9-S10). Here the red shading indicates the minimum to maximum range of the residual bias across grid cells for the base period after bias correction ($E_B$, Eq. (15)), and the solid red line shows the domain average of this bias at individual quantiles concerned. In contrast the grey shading illustrates the minimum to maximum range of the differences in the change between the base and evaluation period of the raw model and observations, respectively ($E_\Delta$, Eq. (16)). The black solid line marks the domain average of this bias at individual quantiles. The residual bias for the evaluation period $E_F$ (or in analogy any other future time period) comprises the sum of these errors (base bias and response bias) and is illustrated for the domain average with the dashed yellow line. We note that, as the DC method by definition yields no initial error in the base period only $E_\Delta$ is relevant in the evaluation period which is illustrated by the grey shading and the black solid line in all panels of Fig. 5 (as well as Fig. S9-S10).

For the base period it is apparent that the QM correction technique, in contrast to RB and MB correction, yields only minor differences across the MDA8 $O_3$ distribution when compared to the observations in both spatial domains. For the evaluation period we see that the difference in response between models and observations is dominating over the raw performance of the individual correction techniques and that the residual bias depends strongly on region and model concerned (see Fig. 4-5 and supplemental Fig. S9-S10). Given this result, we assume that the correction performance depends strongly on models being able to represent precursor emission changes over time as seen in observations.

All models show distinct biases in reproducing observed ozone changes between the two time periods, with a particularly pronounced magnitude in the tails of the distributions. Although both error terms, and the resulting net error are found to be rather small in the domain average (roughly ±5 ppb), they might have a strong influence on the individual grid cell level (see shadings). Especially for the MB and RB techniques the individual errors might compensate each other, as illustrated by the improved results relative to the base period. The DC and QM approaches on the other hand strongly depend on the quality of the model response in time. Here we find, that pronounced errors in the model change offset (at least in part) the benefits illustrated for the base period (see Fig. 4).

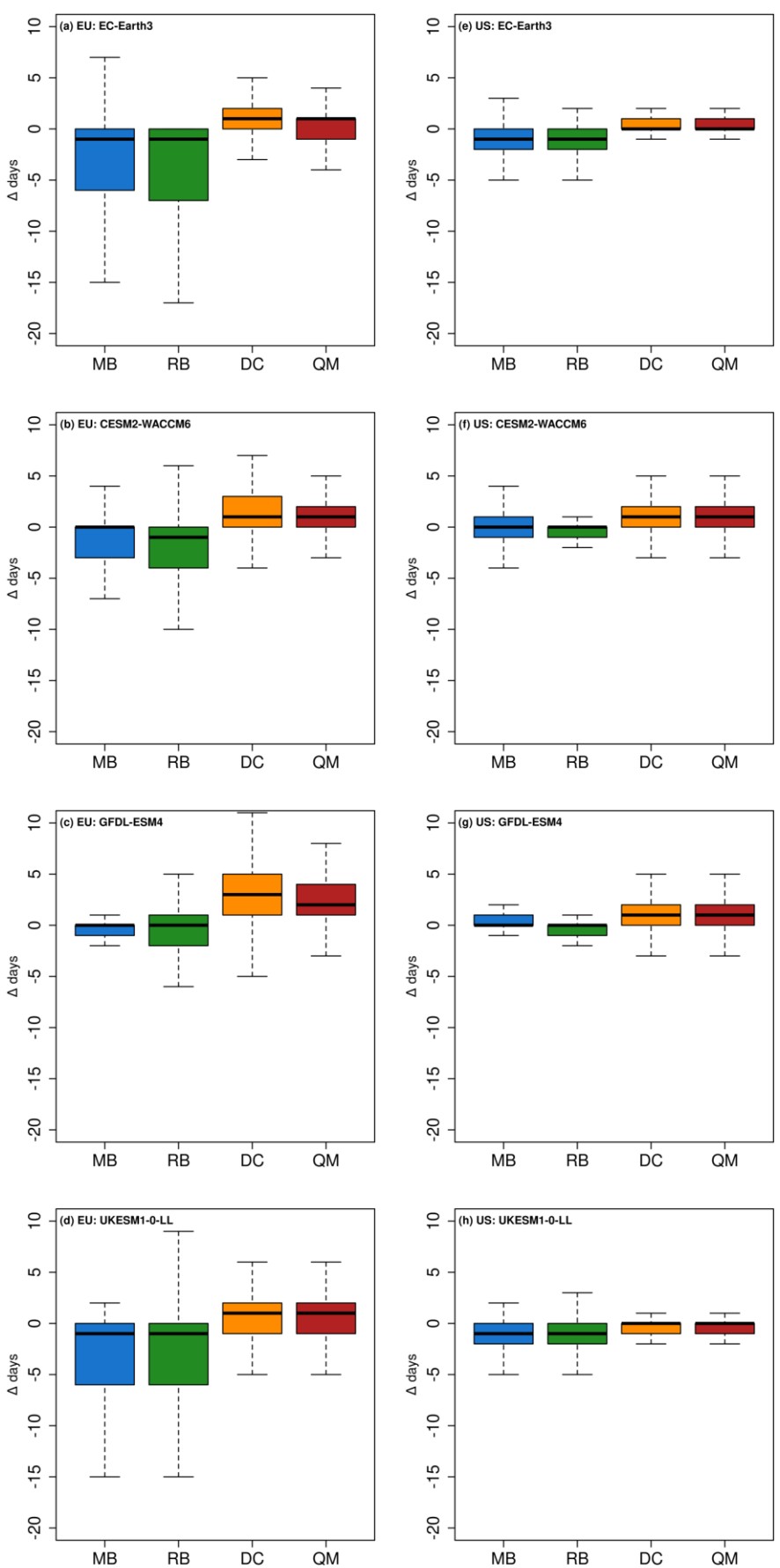

 **Fig 4:** As Fig. 3 but for 2010-2014 time period. Blue, Green, orange and red colour indicates the MB, RB, DC and QM correction methods respectively.

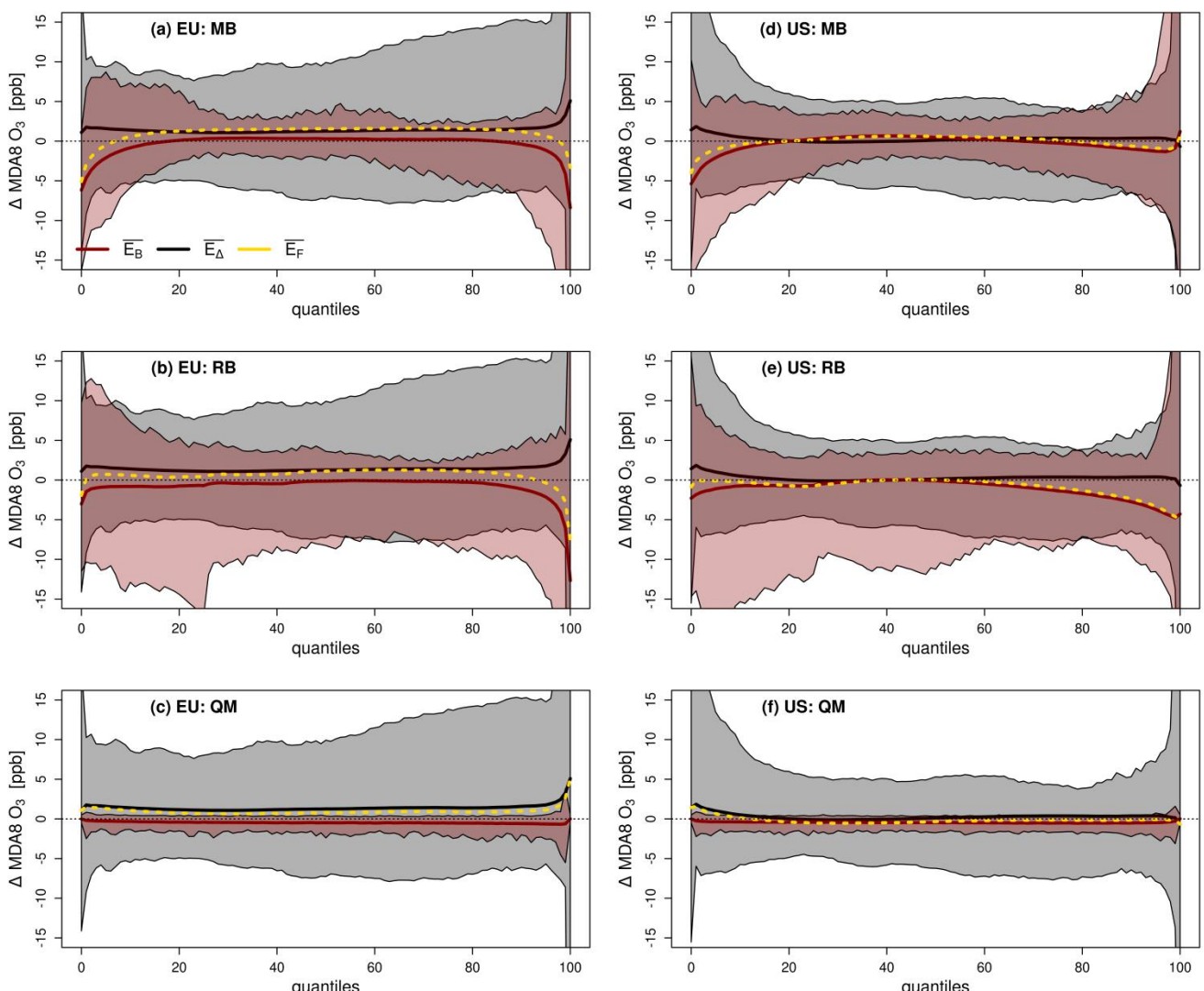

**Fig. 5:** Error Components of the CESM2-WACCM6 model during the evaluation period, for the MB (a & d), RB (b & e) and QM (c & f) methods, in the EU (left column) and US (right column) domain. The red shading gives the minimum to maximum range, and the solid red line the domain average of the residual bias in the base period ($E_B$), respectively. The grey shading gives the minimum to maximum range and the solid black line the domain average of the differences in the change between the base and evaluation period of the raw model and observations ($E_\Delta$), respectively. The resulting domain average error of the evaluation period ($E_F$) is indicated by the dashed yellow line (note, for the DC method $E_B = 0$ and hence $E_F = E_\Delta$).

### 3.4 The influence of meteorology on the bias in the CESM2-WACCM6 ensemble

Having illustrated the MDA8 O$_3$ biases of various CMIP6 models, the performance of various statistical bias techniques as well as the influence of the model response to changes in e.g. emissions on the performance of bias correction we turn here to shed light on the underlying cause of biased MDA8 O$_3$ model outputs. To this end we analyze the 13 members of the CESM2-WACCM6 ensemble in more detail, in order to examine for consistency within the individual realizations as well

as a possible dominant cause(s) for the bias in the modelled surface ozone fields. Here two likely prime candidates exist: 1) issues with the sensitivity in chemical mechanisms to local/regional precursor emissions (note, anthropogenic emissions are consistent across individual models), 2) issues in meteorology simulated by the free running CCM. For the latter, we further include three climatological key drivers for ozone production/accumulation in our analysis, i.e. daily maximum temperature (TSMX), daily average down welling short wave radiation (FSDS) and daily average wind speed (WSPD), in order to differentiate whether the bias is predominantly driven by sensitivity to meteorology or chemistry. As chemical covariates we include monthly averages of NO, $NO_2$ and HCHO, the latter we consider as bulk proxy for VOCs [e.g. Shen et al., 2019; Zhu et al., 2017].

Figures 6 and 7 illustrate the PDFs of MDA8 $O_3$, NO, $NO_2$, HCHO, TSMX, FSDS, and WSPD for the individual ensemble members during spring and summer in 2005-2009 (the PDFs for 2010-2014 are shown in supplemental Fig. S11 and S12). MAM and JJA MDA8 $O_3$ (Fig. 6a, e) show a very similar distribution across ensemble members for both domains. For example the median MDA8 $O_3$ value ranges across ensemble members roughly between 50 and 52 ppb (MAM) and 45 to 47 ppb (JJA) in the EU. For the US the median MDA8 $O_3$ values are found to be slightly higher than in the EU, but the differences within the ensemble lie in the same narrow range (53 to 55 ppb for MAM and 54 to 55 ppb for JJA). Similarly, compact PDFs across the ensemble are found for NO, $NO_2$ and HCHO. Interestingly differences emerge for HCHO in the US but not Europe which represents a larger influence of biogenic emissions.

Similar results are found for the meteorological variables. Although slight variations occur for surface temperature, radiation, and wind speed (which one would expect from a model generating its own meteorology), the PDFs are widely homogenous across the ensemble, thereby explaining the similarity of surface ozone distributions within the ensemble (as all ensemble members are driven with the same set of precursor emissions) in both domains. The analysis of the MDA8 $O_3$, NO, $NO_2$, HCHO, TSMX, FSDS, and WSPD distributions over the second time period (2010-2014, Fig. S11 and S12) yields similar results, thereby providing confidence for the robustness of our findings.

The strong similarity across ensemble members indicates that the MDA8 $O_3$ bias identified in CESM2-WACCM6 stems most likely from sensitivities in the chemical mechanism and/or emissions and not from meteorological drivers and their variability. As the models use the same anthropogenic emissions, the differences are more likely to stem from the chemistry, which could include different mixes of emitted VOCs. Previous research has shown that temperature biases are rather small and that a significant overestimation of the temperature in the troposphere solely occurs in the southern hemisphere polar region, a region which is not investigated here [Danabasoglu et al., 2020; Gettelman et al., 2019]. Nevertheless, we note that small deviations in temperature have been found to explain biases of 5-15 ppb for surface $O_3$ in former model generations [Rasmussen et al., 2012]. While the presented ensemble analysis is, due to data availability, only possible for CESM2-WACCM6, the results provide a first order estimate of the dominant model component responsible for surface ozone biases. Future work should confirm that this finding holds for other global models and thus an ensemble strategy for model experiments is recommended for future model intercomparison activities such as CCMI and CMIP.

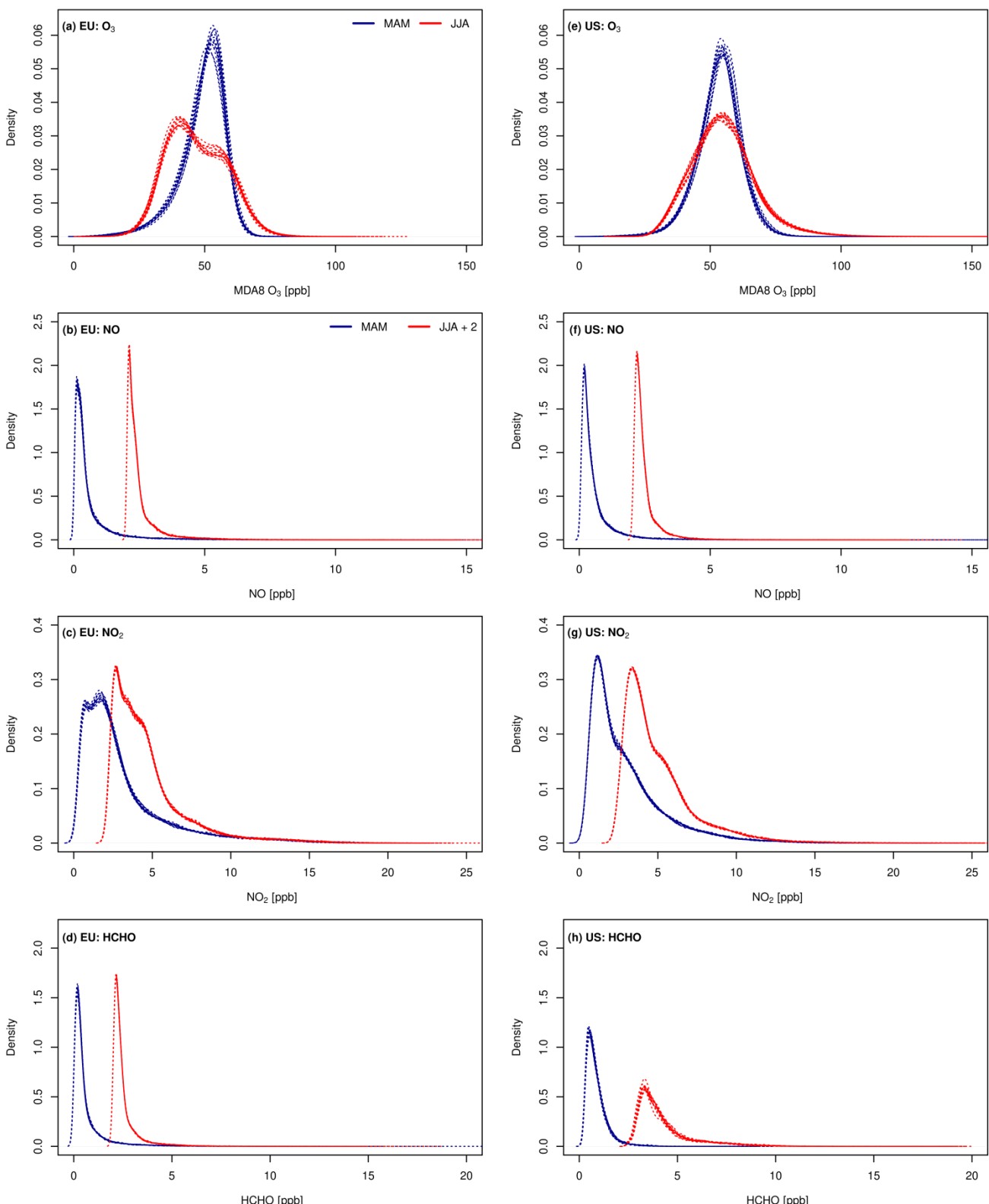

**Fig. 6:** CESM2-WACCM6 spring (MAM) and summer time (JJA) PDFs of (a) MDA8 $O_3$, (b) NO, (c) $NO_2$, (d) HCHO for the Europeandomain in 2005-2009. (e)-(h) as (a)-(b) but for the US domain Note, a value of 2 has been added to summertime concentrations of NO, $NO_2$ and HCHO to allow for visual separation of the seasonal PDFs.

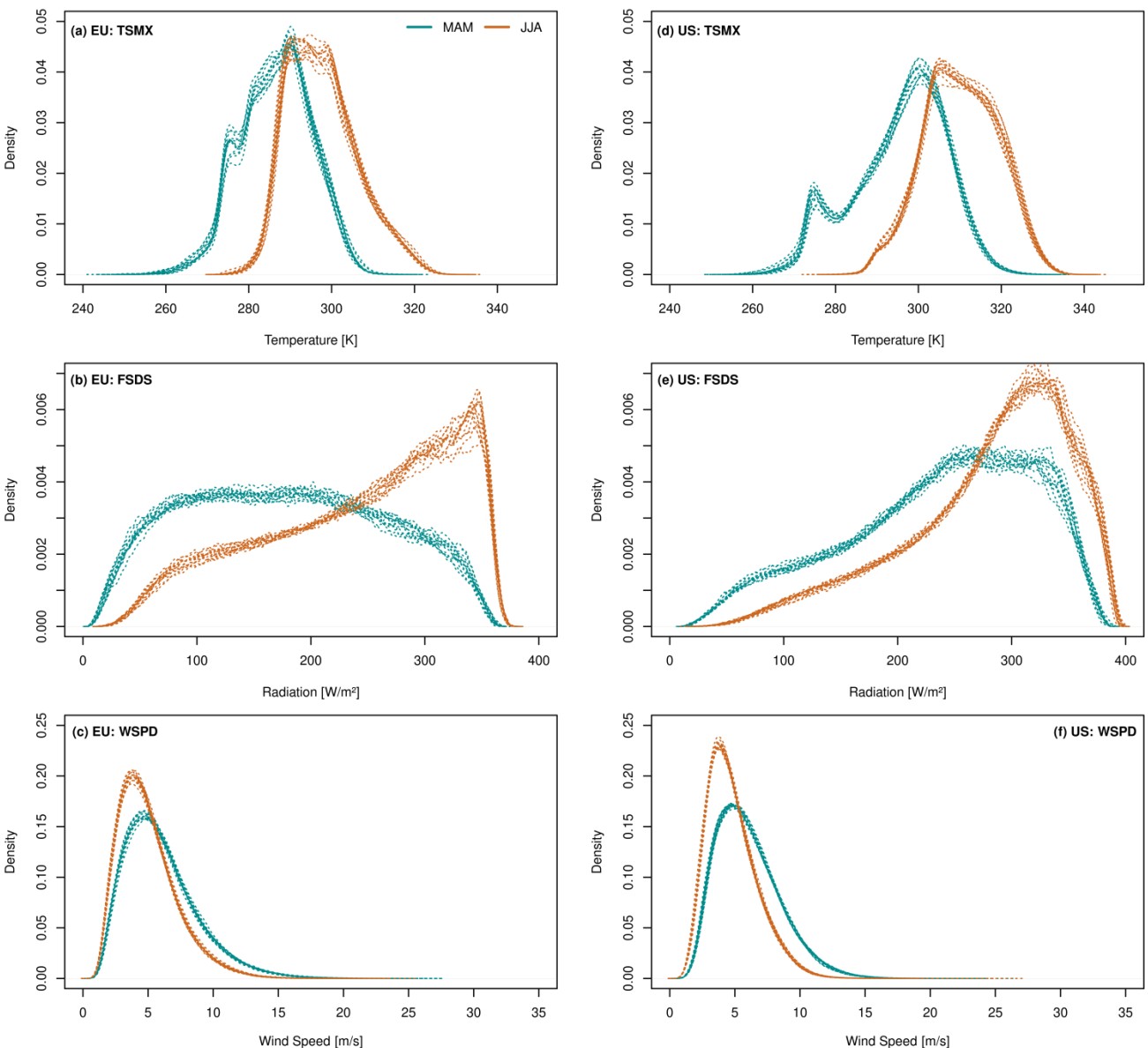

**Fig. 7:** CESM2-WACCM6 spring (MAM) and summer time (JJA) PDFs of (a) TSMX, (b) FSDS, and (c) WSPD for the European domain in 2005-2009. (d)-(f) as (a)-(c) but for the US domain.

## 4 Summary & Conclusions

In this study we evaluate four global CCMs contributing to CMIP6 (EC-Earth3, CESM2-WACCM6, GFDL-ESM4 and UKESM1-0-LL) regarding their bias in surface ozone burdens, and present the first comprehensive comparison of the performance of four different statistical bias correction techniques to derive CCM-based ozone metrics with relevance for public health and policy. While all models show biases when compared to observations, the bias magnitude of the raw, un-corrected MDA8 $O_3$ outputs differs strongly within the pool of models analyzed.

The evaluation of the four bias correction techniques for the base period (2005-2009), where techniques are tuned to observations, illustrates that all methods are capable to lower the bias. The MB and RB methods, however, are less accurate when contrasted with the results obtained with the DC or QM approaches. Furthermore, when applying the MB and RB

methods the model output fields might even be overcorrected for individual grid cells, i.e., the resulting ozone distributions might become biased low. This is not surprising as both techniques apply a single average value for the correction of the whole distribution function, which is a disadvantage - especially when it comes to the tails of the distribution - if the bias is not constant across the ECDF.

The independent evaluation of the four techniques over the second time period (2010-2014), focusing on the bias correction of model projections, yields less distinct results. Although the model-to-observation agreement is improved for all MDA8 $O_3$ metrics in the corrected models compared to their raw counterparts, no single optimal correction technique can be identified. Our results illustrate that technique performance depends strongly on the model selected and its MDA8 $O_3$ evolution, and thus response to boundary condition changes, over time. This at first surprising result, however, can be explained by the examination of the composition of the residual model error.

The residual error for future projections is comprised of two parts: 1) the residual error of the base period $E_B$, and 2) the error attributable to the model response to changes in boundary conditions (emissions, climate, etc…) between both time periods $E_\Delta$. The magnitude of $E_\Delta$ was found to exert a dominant influence on the overall correction performance, which raises some concerns regarding the robustness of model responses and thus the reliability of model projections (not only in context of surface $O_3$). In contrast to $E_\Delta$, $E_B$ depends on the quality of the initial base period bias correction. Here our results clearly show that $E_B$ is substantially larger for the MB and RB than the QM and DC methods. When applying the correction techniques $E_\Delta$ and $E_B$ might compensate for individual grid cells, resulting in a low residual bias. On the contrary, the strong performance for the base period obtained with the QM and DC approaches are attributable to a very low $E_B$, which might deteriorate in projections if $E_\Delta$ is large. Thus, we conclude, that under the assumption of an adequate model response to changing boundary conditions (and thus low $E_\Delta$), the QM and DC methods are outperforming the MB and RB techniques. If a decision has to be made whether the DC or QM approach is used for bias correction, we would argue, given that differences between the results obtained with both techniques are negligible, for DC correction due the comparably easy numerical implementation.

To obtain further insights of the root cause(s) of the bias surface ozone in models, we explored the MDA8 $O_3$ output of the 13 member CESM2-WACCM6 ensemble together with information on NO, $NO_2$ and HCHO as well as key meteorological covariates for ozone production, i.e. daily maximum temperature, daily mean incoming shortwave radiation and daily mean wind speed. Here our analysis showed only small variations within the CESM2-WACCM6 ensemble for core meteorological drivers (and chemical covariates) of surface ozone. This suggests, given that emissions are consistent across models, a dominant influence of the chemical mechanism on the bias in the $O_3$ fields rather than a prominent role for model meteorology. To investigate if this finding can be generalized to other CCMs requires future community efforts in the provision of additional ensemble simulations for individual CCMs contributing to the CCMI or CMIP frameworks.

**Data availability**

CMIP6 data sets are publicly available via https://esgf-data.dkrz.de/projects/cmip6-dkrz/. Processed data can be made available by the corresponding author upon reasonable request. The gridded MDA8 $O_3$ datasets are available at: https://doi.org/10.5281/zenodo.10832955

**Author Contribution**

C. Staehle: conceptualization, formal analysis, methodology, visualization, writing – original draft preparation

H. E. Rieder: conceptualization, methodology, resources, supervision, writing – review and editing

A. M. Fiore: resources, supervision, writing – review and editing

J. Schnell: data curation, writing – review and editing

**Competing Interests**

The authors declare that they have no conflict of interest.

**Acknowledgement**

The authors are grateful to the EC-Earth3, GFDL-ESM4 and UKESM1-0-LL modelling teams for providing the ensemble simulations via https://esgf-data.dkrz.de/projects/cmip6-dkrz/. H.E. Rieder and C. Staehle acknowledge partial support by the Austrian Climate and Energy Fund via project ACRP11-KR18AC0K14686. C. Staehle acknowledges support through an OeAD Marietta Blau Fellowship grant. This research was supported in part by the NOAA cooperative agreement NA22OAR4320151, for the Cooperative Institute for Earth System Research and Data Science (CIESRDS). The statements, findings, conclusions, and recommendations are those of the author(s) and do not necessarily reflect the views of NOAA or the U.S. Department of Commerce. The authors are grateful to Ramiro Checa-Garcia for fruitful discussions and comments. The authors thank two anonymous reviewers for their valuable comments on an earlier version of this manuscript.

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
