# Peer review of "Technical note: An assessment of the performance of statistical bias correction techniques for global chemistry-climate model surface ozone fields"

_EGUsphere, 2023_

## Author Response (AR1)

We thank two anonymous reviewers for their constructive comments. Below we provide a point by point response (*blue, italics*) to the individual reviewer comments (black) and specific changes made to the manuscript (*red, italics*).

**Reviewer 1:**

This is an interesting study on different bias-correction methods applied to CMIP6 Earth System Models (ESMs) surface-ozone results. The analysis explores the performance of each applied statistical bias-correction method, to what extent this is sensitive to each individual ESM, and finally the nature and origin of the errors. The manuscript is well organized with qualitative and efficient presentation of the results. Yet, there are some points that need clarification and further investigation. I believe the study may be a valuable addition to the literature once the following comments are addressed.

*We thank the reviewer for the positive assessment of our manuscript and the valuable comments provided that helped strengthen our study. Please find our point by point response to the individual comments below.*

**Main Comments**

1. More information on the gridded observational ozone dataset (used here for the evaluation) is needed. Do the authors use the Schnell et al. (2014) data? This assessment is for the 2000-2009 period. Is this an extension of this dataset? Is this dataset publicly available? Please describe (briefly) in the manuscript how this dataset was constructed. Is the inhomogeneous network of observations over Europe and USA affecting the results of your evaluation and how? This should be discussed. I suggest including a subsection in the Data & Methods Section about the gridded observational ozone data and relevant information.

*We thank the reviewer for this comment. We use an updated and extended version of the data set presented in Schnell et al. (2014). The data set comprises surface ozone observations from monitoring networks in Europe and the US. Precisely, for the US the data set is based on the EPA's Air Quality System (AQS) and Clean Air Status and Trends Network (CASTNET), and Environment Canada's National Air Pollution Surveillance Program (NAPS), while for Europe the data set combines the EMEP and the European Environment Agency's AirBase network (excluding stations designated as traffic). Records from EMEP and AirBase are reported as µg m$^{-3}$ and are converted to parts per billion (ppb = $10^{-9}$ mol mol$^{-1}$ = nmol mol$^{-1}$) using a temperature of 20 °C. The gridded product is created following the procedures described in Section 2 of Schnell et al. (2014). We have included the following additional text in the revised manuscript:*

*We also obtain observed MDA8 $O_3$ with a spatial resolution of 1° x 1° per grid cell for both the European and the US domain using an extended dataset constructed using the methods of Schnell et al. [2014; 2015] and Schnell and Prather [2017], one which was designed specifically to compare against gridded CCMs. The dataset is constructed using an inverse distance weighted interpolation method that includes a declustering component similar to kriging; i.e., clustered (within 100 km) observations' weights are reduced such that those stations (often located around urban centers) are not disproportionately used in the interpolation. For the US domain, point based observations that are used in the interpolation include*

*the US EPA's Air Quality System (AQS), the US EPA Clean Air Status and Trends Network (CASTNET), and Environment Canada's National Air Pollution Surveillance Program (NAPS); for the European Domain we include the EMEP and European Environment Agency's AirBase network (excluding stations designated as traffic). The exponent for the distance component is 2.5 and a maximum distance of 500 km is used for the weights. Parameters were estimated using a leave-N-out cross-validation technique. Estimations are made at 25 equally spaced points within each 1° x 1° cell and trapezoidally averaged. Other recent work has used this extended dataset [e.g., Ducker et al., 2018; Garrido-Perez et al., 2019, Guo et al, 2018]. Schnell et al. [2014] estimated an RMSE of 6-9 ppb for individual stations and 0-3 ppb for the grid cell averages; Ducker et al. [2018] estimated a mean bias of 5-10 ppb with the updated dataset over their study locations. For the analysis here the interpolation is performed on hourly abundances and the MDA8 $O_3$ is estimated using the interpolated hourly fields. Note, we apply the nomenclature of the European Union for the calculation of the MDA8 $O_3$ values in both domains, i.e. the eight hour average for a given hour is derived using the data of that specific hour and the preceding seven hours [EUR-LEX, 2008]. For convenience, the data is provided along with this article, see data statement below. To allow for an optimal comparison, the model data is regridded using an ordinary inverse distance weighting algorithm to match the spatial extent of the observations.*

2. To explore the error sources, the authors select the daily maximum temperature and radiation for sensitivities to meteorology. Yet, wind and especially for high-ozone events stability (stagnation) are also important drivers. How are these two represented by the individual ensemble members? Attributing model error mainly to precursors emissions needs more evidence. What are the NOx and VOC PDFs for the ensemble members? Are there any model diagnostics for ozone production (PO3) and loss (PO3) to support this?

*We thank the reviewer for this comment. Following the reviewer's suggestion we have included additional meteorological and chemical variables available from the CESM2-WACCM6 simulations in the analysis. These are daily mean wind speed, monthly mean concentrations of $NO_2$, NO and HCHO (as VOC proxy). We detail this expanded analysis now in our but only a single ensemble member. Therefore, these metrics could not be included in the analysis here. manuscript in the revised section 3.4. We agree with the reviewer that ozone production and ozone loss would be useful terms, however those are unfortunately not available (as not archived) for the ensemble*

*We updated section 3.4. as provided below:*

*Having illustrated the MDA8 $O_3$ biases of various CMIP6 models, the performance of various statistical bias techniques as well as the influence of the model response to changes in e.g. emissions on the performance of bias correction we turn here to shed light on the underlying cause of biased MDA8 $O_3$ model outputs. To this end we analyse the 13 members of the CESM2-WACCM6 ensemble in more detail, in order to examine for consistency within the individual realizations as well as a possible dominant cause(s) for the bias in the modelled surface ozone fields. Here two likely prime candidates exist: 1) issues with the sensitivity in chemical mechanisms to local/regional precursor emissions (note, anthropogenic emissions are consistent across individual models), 2) issues in meteorology simulated by the free running CCM. For the latter, we further include three climatological key drivers for ozone production/accumulation in our analysis, i.e. daily maximum temperature (TSMX), daily average down welling short wave radiation (FSDS) and daily average wind speed (WSPD), in order to differentiate*

*whether the bias is predominantly driven by sensitivity to meteorology or chemistry. As chemical covariates we include monthly averages of NO, NO$_2$, HCHO, the latter we consider as bulk proxy for VOCs [e.g. Shen et al., 2019; Zhu et al., 2017].*

*Figures 6 and 7 illustrate the PDFs of MDA8 O$_3$, NO, NO$_2$, HCHO, TSMX, FSDS, and WSPD for the individual ensemble members during spring and summer in 2005-2009 (the PDFs for 2010-2014 are shown in the supplemental Fig. S11 and 12). MAM and JJA MDA8 O$_3$ (Fig. 6a,e) show a very similar distribution across ensemble members for both domains. For example the median MDA8 O$_3$ value ranges across ensemble members roughly between 50 and 52 ppb (MAM) and 45 to 47 ppb (JJA) in the EU. For the US the median MDA8 O$_3$ values are found to be slightly higher than in the EU, but the differences within the ensemble lie in the same narrow range (53 to 55 ppb for MAM and 54 to 55 ppb for JJA). Similarly, compact PDFs across the ensemble are found for NO, NO$_2$ and HCHO. Interestingly differences emerge for HCHO in the US but not Europe which represents a larger influence of biogenic emissions.*

*Similar results are found for the meteorological variables. Although slight variations occur for surface temperature radiation, and wind speed (which one would expect from a model generating its own meteorology), the PDFs are widely homogenous across the ensemble, thereby explaining the similarity of surface ozone distributions within the ensemble (as all ensemble members are driven with the same set of precursor emissions) in both domains. The analysis of the MDA8 O$_3$, NO, NO$_2$, HCHO, TSMX, FSDS, and WSPD distributions over the second time period (2010-2014, Figs. S11 and S12) yields similar results, thereby providing confidence for the robustness of our findings.*

*The strong similarity across ensemble members indicates that the MDA8 O$_3$ bias identified in CESM2-WACCM6 stems most likely from sensitivities in the chemical mechanism and/or emissions and not from meteorological drivers and their variability. As the models use the same anthropogenic emissions, the differences are more likely to stem from the chemistry, which could include different mixes of emitted VOCs….*

3. It would be interesting to see results for MDA8 O$_3$ using a different gridded observational ozone dataset (if available). Moreover, since the ultimate purpose of the study is to support reliability of ozone-health studies, the recent Global Burden of Disease (GBD) report (2019) applies the ozone season daily maximum 8 hour mixing ratio (OSDMA8) metric to estimate excess mortality from long-term ozone exposure. Gridded observational OSDMA8 data, as described in DeLang et al. (2021), are publicly available at https://zenodo.org/records/8320001. Are the statistical methods used here applicable for a long-term effect ozone metric like OSMDA8?

*Following the reviewer's suggestion we have expanded the analysis towards OSDMA8. To this end we have computed OSDMA8 from our data set following the method described in DeLang et al. (2021).*

*Our results show a varying OSDMA8 bias across CCMs for the periods 2005-2009 and 2010-2014 (see Figure R1). This is in agreement with the bias in MDA8 O$_3$ we report on. As for MDA8 O$_3$, after applying individual correction techniques the bias is substantially reduced (see Figures R2 and R3). For OSDMA8 however, the individual bias correction techniques (despite all being able to substantially reduce the bias) show no pronounced difference in performance. We attribute this to properties of the OSDMA8 metric itself, which represents the annual maximum value of the running 6-month average of monthly mean MDA8 O$_3$, i.e., a single value per calendar year. Given the long-term averaging time span considered for the OSDMA metric we do expect also no major difference among techniques explored here. More*

*illustrative for the performance of the correction techniques across the full range of the ozone burden is the evaluation of the residual bias in the MDA8 O₃ distribution function explored in Fig. S5 and S8 in our original manuscript. Therefore, for clarity and focus of the present study we prefer to restrict the analysis to MDA8 O₃.*

[Figure]

*Fig. R1: Boxplots of the residual bias in OSDMA8 pooled across grid cells in the individual CCMs in 2005-2009 and 2010-2014 for the EU (a,b) and US domain (c,d).*

[Figure]

*Fig. R2: Boxplots of the residual bias in OSDMA pooled across grid cells for individually bias corrected CCMs in 2005-2009 for the EU (a,b,c,d) and US domain (e,f,g,h). Blue, Green and red colour indicate the MB, RB and QM correction methods respectively.*

[Figure]

*Fig. R3: as Fig. R2 but for 2010-2014.*

**Comments**

L18-19: This is a strong statement as this is not explicitly shown from the results. See also main comment #2.

*Following the suggestion of the referee, we have included further available meteorological and chemical covariates in the analysis. These show all strong uniformity across the CESM2-WACCM6 ensemble, confirming our original postulated hypothesis. However, as we agree with the referee that we cannot show the root cause of model bias in full explicity, and only have a large ensemble for one global model available, we further have included a limiting qualifier in this abstract statement:*

*Ensemble simulations available for one CCM indicate that model ozone biases are likely more sensitive to the process representation embedded in chemical mechanisms or emissions rather than to meteorology.*

L21-26: Tropospheric and therefore surface ozone has also a natural source, the transport from the stratosphere (Stohl et al., 2003) which over specific regions (Lin et al., 2015) or occasionally (Akritidis et al., 2010) contributes significantly.

*Thank you for this comment. We have updated the introduction section respectively.*

*Tropospheric $O_3$ abundance is also substantially influenced by stratospheric intrusions, which can in certain regions or during specific events alter concentrations significantly [Akritidis et al., 2010; Lin et al., 2015; Stohl et al., 2003].*

L26: "O3 is associated with a variety of detrimental human health effects". I suggest including here a couple of recent references on ozone effects on human health like Murray et al. (2020) and Pozzer et al. (2023).

*Thank you for this comment. We have included several references regarding health effects of ozone.*

*$O_3$ is associated with a variety of detrimental human health effects, especially in the context of the respiratory and cardiovascular system, resulting in about 5-20 % of premature deaths attributable to ambient air pollution [Gu et al., 2023; Malashock et al., 2022; Monks et al., 2015; Murray et al., 2020; Pozzer et al., 2023; Zhang et al., 2019].*

L40: maybe "meteorology and deposition"

*Thank you, done.*

L65-66: Please clarify why only the first member of the ensemble is used in the main study.

*We have included the clarification in the revised manuscript. Please see also our response to comment 4 of Reviewer 2.*

*For most of our study, we use only the first ensemble member of CESM2-WACCM6 in analogy to the other CCMs given the overall heterogeneity in the number of members available per model. In section*

*4.3, we focus on the chemical vs. meteorological driving of model biases and utilize the entire CESM2-WACCM6 ensemble.*

L68: The period 1993 to 2014 is referred here. Are there any data used in the analysis except from the 2005-2009 and 2010-2014 periods? Please clarify.

*No observational data beyond 2005-2014 has been used in our study. Note, we have overall revised section 2.1 following the comments provided by both referees.*

L183: Remove t.

*Thank you for spotting this typo.*

L285-286: As this is not explicitly shown to be related with precursors emissions but rather assumed I suggest rephrasing accordingly.

*We have revised this sentence accordingly.*

*Given this result, we assume that the correction performance depends strongly on models being able to represent precursor emission changes over time as seen in observations.*

L360: $E_B$ depends

*Thank you, done.*

L361: $E_B$ is

*Thank you, done.*

L363: "the strong base period performance", maybe "the strong performance for the base period"?

*We have revised this sentence accordingly.*

*On the contrary, the strong performance for the base period obtained with the QM and DC approaches are attributable to a very low $E_B$, which might deteriorate in projections if $E_\Delta$ is large.*

**References**

Akritidis D., P. Zanis, I. Pytharoulis, A. Mavrakis and Th. Karacostas,: A deep stratospheric intrusion event down to the earth's surface of the megacity of Athens, Meteorology and Atmospheric Physics, 109 (1-2), 9-18, DOI: 10.1007/s00703-010-0096-6, 2010

DeLang MN, Becker JS, Chang KL, Serre ML, Cooper OR, Schultz MG, et al. Mapping Yearly Fine Resolution Global Surface Ozone through the Bayesian Maximum Entropy Data Fusion of Observations and Model Output for 1990–2017. Environ Sci Technol [Internet]. 2021 Apr 20;55(8):4389–98. Available from: https://doi.org/10.1021/acs.est.0c07742

Lin, M., Fiore, A., Horowitz, L. et al. Climate variability modulates western US ozone air quality in spring via deep stratospheric intrusions. Nat Commun 6, 7105 (2015). https://doi.org/10.1038/ncomms8105

Murray CJL, Aravkin AY, Zheng P, Abbafati C, Abbas KM, Abbasi-Kangevari M, et al. Global burden of 87 risk factors in 204 countries and territories, 1990–2019: a systematic analysis for the Global Burden of Disease Study 2019. The Lancet [Internet]. 2020;396(10258):1223–49. Available from: https://www.sciencedirect.com/science/article/pii/S0140673620307522

Pozzer A, Anenberg SC, Dey S, Haines A, Lelieveld J, Chowdhury S. Mortality Attributable to Ambient Air Pollution: A Review of Global Estimates. Geohealth [Internet]. 2023;7(1):e2022GH000711. Available from: https://agupubs.onlinelibrary.wiley.com/doi/abs/10.1029/2022GH000711

Stohl, A., et al. (2003), Stratosphere-troposphere exchange: A review, and what we have learned from STACCATO, J. Geophys. Res., 108, 8516, doi:10.1029/2002JD002490, D12.

*Thank you for providing these valuable references, we have included these and others in the revised version of our manuscript.*

**References added beyond those suggested by the referee**

*Ducker, J. A., C. D. Holmes, T. Keenan, S. Fares, A. Goldstein, I. Mammarella, W. Munger, and J. L. Schnell (2018), Synthetic ozone deposition and stomatal uptake at flux tower sites, Biogeosci., 15, 5395-5413, https://doi.org/10.5194/bg-15-5395-2018.*

*EUR-LEX. (2008). Directive 2008/50/EC of the European parliament and of the council of 21 May 2008 on ambient air quality and cleaner air for Europe (2008/50/EC). Retrieved from https://eur-lex.europa.eu/legal-content/EN/TXT/?uri=CELEX:32008L0050*

*Garrido-Perez, J. M., C. Ordóñez, R. García-Herrera, and J. L. Schnell (2019), The differing impact of air stagnation on summer ozone across Europe, Atmos. Environ., 219, 117062, https://doi.org/10.1016/j.atmosenv.2019.117062.*

*Guo, J, A. M. Fiore, L. T. Murray, D. A. Jaffe, J. L. Schnell, T. Moore, and G. Milly (2018), Average versus high surface ozone levels over the Continental U.S.A.: Model bias, background influences, and interannual variability, Atmos. Chem. Phys., 18, 12123-12140, https://doi.org/10.5194/acp-18-12123-2018.*

*Schnell, J. L., M. J. Prather, B. Josse, V. Naik, L. W. Horowitz, P. Cameron-Smith, D. Bergmann, G. Zeng, D. A. Plummer, K. Sudo, T. Nagashima, D. T. Shindell, G. Faluvegi, and S. A. Strode (2015), Use of North American and European air quality networks to evaluate global chemistry-climate modeling of surface ozone, Atmos. Chem. Phys., 15(18), 10581-10596, https://doi.org/10.5194/acp-15-10581-2015*

*Schnell, J. L., Prather, M. J. (2017). Co-occurrence of extremes in surface ozone, particulate matter, and temperature over eastern North America. Proceedings of the National Academy of Sciences, 114(11), 2854-2859. https://doi.org/10.1073/pnas.1614453114*

**Reviewer 2:**

The study presents different bias correction methods for surface ozone and apply them to four CMIP6generation Earth System Models (ESMs). The performance of each applied method is investigated along with the sensitivities of each individual ESM to these methods, and finally recommendations. The manuscript is well-organized and easy to follow. There are few points that need further clarification before it can be accepted in ACP.

*We thank the reviewer for the positive assessment of our manuscript and the valuable comments provided that helped strengthen our study. Please find our point by point response to the individual comments below.*

**General comments**

1. More information is needed on the gridded observational ozone dataset, including how this dataset was generated briefly, referring to the observation networks in Europe and USA.

*Thank you for this comment. Following your comment and the comments of Ref #1 we have included the text below in the revised manuscript.*

*We also obtain observed MDA8 $O_3$ with a spatial resolution of 1° x 1° per grid cell for both the European and the US domain using an extended dataset constructed using the methods of Schnell et al. [2014; 2015] and Schnell and Prather [2017], one which was designed specifically to compare against gridded CCMs. The dataset is constructed using an inverse distance weighted interpolation method that includes a declustering component similar to kriging; i.e., clustered (within 100 km) observations' weights are reduced such that those stations (often located around urban centers) are not disproportionately used in the interpolation. For the US domain, point based observations that are used in the interpolation include the US EPA's Air Quality System (AQS), the US EPA Clean Air Status and Trends Network (CASTNET), and Environment Canada's National Air Pollution Surveillance Program (NAPS); for the European Domain we include the EMEP and European Environment Agency's AirBase network (excluding stations designated as traffic). The exponent for the distance component is 2.5 and a maximum distance of 500 km is used for the weights. Parameters were estimated using a leave-N-out cross-validation technique. Estimations are made at 25 equally spaced points within each 1° x 1° cell and trapezoidally averaged. Other recent work has used this extended dataset [e.g., Ducker et al., 2018; Garrido-Perez et al., 2019, Guo et al, 2018]. Schnell et al. [2014] estimated an RMSE of 6-9 ppb for individual stations and 0-3 ppb for the grid cell averages; Ducker et al. [2018] estimated a mean bias of 5-10 ppb with the updated dataset over their study locations. For the analysis here the interpolation is performed on hourly abundances and the MDA8 $O_3$ is estimated using the interpolated hourly fields. Note, we apply the nomenclature of the European Union for the calculation of the MDA8 $O_3$ values in both domains, i.e. the eight hour average for a given hour is derived using the data of that specific hour and the preceding seven hours [EUR-LEX, 2008]. For convenience, the data is provided along with this article, see data statement below. To allow for an optimal comparison, the model data is regridded using an ordinary inverse distance weighting algorithm to match the spatial extent of the observations.*

2. The dataset is divided into two for evaluation and projections. It would be useful to show if projections would give similar results if other datasets would be used, or the projections would be applied in other regions such as Asia. In addition, would the conclusions change if another metric was used to evaluate the performance.

*We agree with the reviewer that expanding to additional data sets or study regions would be of interest. However, we consider this beyond the scope of the present work. We have chosen to restrict the analysis here to the data product included here given that 1) to the best of our knowledge no other gridded observational data set for MDA8 $O_3$ is freely available for multiple other world regions and/or created across regions with a uniform methodological framework; and 2) no other consistent gridded data products for MDA8 $O_3$ are available for both the EU and the US.*

3. Daily maximum temperature and radiation are selected for sensitivity to meteorology. I would recommend looking at winds to account for transport. Is there a reason why it is not included? Another important source is stratospheric ozone, which should be discussed.

*We thank the reviewer for this comment. Following your and reviewer 1's suggestion we have included additional variables, including information on mean wind speed, and also NO, $NO_2$ and HCHO in the analysis focusing on meteorological vs. chemical driving of model biases. We detail this expanded analysis now in our manuscript in the revised section 3.4.*

*Having illustrated the MDA8 $O_3$ biases of various CMIP6 models, the performance of various statistical bias techniques as well as the influence of the model response to changes in e.g. emissions on the performance of bias correction we turn here to shed light on the underlying cause of biased MDA8 $O_3$ model outputs. To this end we analyse the 13 members of the CESM2-WACCM6 ensemble in more detail, in order to examine for consistency within the individual realizations as well as a possible dominant cause(s) for the bias in the modelled surface ozone fields. Here two likely prime candidates exist: 1) issues with the sensitivity in chemical mechanisms to local/regional precursor emissions (note, anthropogenic emissions are consistent across individual models), 2) issues in meteorology simulated by the free running CCM. For the latter, we further include three climatological key drivers for ozone production/accumulation in our analysis, i.e. daily maximum temperature (TSMX), daily average down welling short wave radiation (FSDS) and daily average wind speed (WSPD), in order to differentiate whether the bias is predominantly driven by sensitivity to meteorology or chemistry. As chemical covariates we include monthly averages of NO, $NO_2$, HCHO, the latter we consider as bulk proxy for VOCs [e.g. Shen et al., 2019; Zhu et al., 2017].*
*Figures 6 and 7 illustrate the PDFs of MDA8 $O_3$, NO, $NO_2$, HCHO, TSMX, FSDS, and WSPD for the individual ensemble members during spring and summer in 2005-2009 (the PDFs for 2010-2014 are shown in the supplemental Fig. S11 and 12). MAM and JJA MDA8 $O_3$ (Fig. 6a,e) show a very similar distribution across ensemble members for both domains. For example the median MDA8 $O_3$ value ranges across ensemble members roughly between 50 and 52 ppb (MAM) and 45 to 47 ppb (JJA) in the EU. For the US the median MDA8 $O_3$ values are found to be slightly higher than in the EU, but the differences within the ensemble lie in the same narrow range (53 to 55 ppb for MAM and 54 to 55 ppb for JJA). Similarly, compact PDFs across the ensemble are found for NO, $NO_2$ and HCHO. Interestingly differences emerge for HCHO in the US but not Europe which represents a larger influence of biogenic emissions.*

*Similar results are found for the meteorological variables. Although slight variations occur for surface temperature radiation, and wind speed (which one would expect from a model generating its own meteorology), the PDFs are widely homogenous across the ensemble, thereby explaining the similarity of surface ozone distributions within the ensemble (as all ensemble members are driven with the same set of precursor emissions) in both domains. The analysis of the MDA8 $O_3$, NO, $NO_2$, HCHO, TSMX, FSDS, and WSPD distributions over the second time period (2010-2014, Figs. S11 and S12) yields similar results, thereby providing confidence for the robustness of our findings.*

*The strong similarity across ensemble members indicates that the MDA8 $O_3$ bias identified in CESM2-WACCM6 stems most likely from sensitivities in the chemical mechanism and/or emissions and not from meteorological drivers and their variability. As the models use the same anthropogenic emissions, the differences are more likely to stem from the chemistry, which could include different mixes of emitted VOCs....*

4. Why did you only use the first member of the ensemble. This should be clarified and justified.

*In the main body of the manuscript we explicitly analyze only the first ensemble member, in analogy to the other CCMs, where also one ensemble member has been used (driven by inhomogeneous availability of members per model). In section 4.3, focusing on the chemical vs. meteorological driving of model bias we include the entire CESM2-WACCM6 ensemble, comprising 13 members. We provide this clarifying information in the revised manuscript.*

*For most of our study, we use only the first ensemble member of CESM2-WACCM6 in analogy to the other CCMs given the overall heterogeneity in the number of members available per model. In section 4.3, we focus on the chemical vs. meteorological driving of model biases and utilize the entire CESM2-WACCM6 ensemble.*